# Combined Effects of Acrylamide and Ochratoxin A on the Intestinal Barrier in Caco-2 Cells

**DOI:** 10.3390/foods12061318

**Published:** 2023-03-20

**Authors:** Dan Su, Jiawen Lu, Chunchao Nie, Ziyan Guo, Chang Li, Qiang Yu, Jianhua Xie, Yi Chen

**Affiliations:** State Key Laboratory of Food Science and Technology, Nanchang University, 235 Nanjing East Road, Nanchang 330047, China

**Keywords:** acrylamide, ochratoxin A, Caco-2 cells, tight junction, interactive effects

## Abstract

Acrylamide (AA) and ochratoxin A (OTA) are contaminants that co-exist in the same foods, and may create a serious threat to human health. However, the combined effects of AA and OTA on intestinal epithelial cells remain unclear. The purpose of this research was to investigate the effects of AA and OTA individually and collectively on Caco-2 cells. The results showed that AA and OTA significantly inhibited Caco-2 cell viability in a concentration- and time-dependent manner, decreased transepithelial electrical resistance (TEER) values, and increased the lucifer yellow (LY) permeabilization, lactate dehydrogenase (LDH) release and reactive oxygen species (ROS) levels. In addition, the levels of IL-1β, IL-6, and TNF-α increased, while the levels of IL-10 decreased after AA and OTA treatment. Western blot analysis revealed that AA and OTA damaged the intestinal barrier by reducing the expression of the tight junction (TJ) protein. The collective effects of AA and OTA exhibited enhanced toxicity compared to either single compound and, for most of the intestinal barrier function indicators, AA and OTA combined exposure tended to produce synergistic toxicity to Caco-2 cells. Overall, this research suggests the possibility of toxic reactions arising from the interaction of toxic substances present in foodstuffs with those produced during processing.

## 1. Introduction

Acrylamide (AA) is one of the food toxins associated with processing and has been detected in foods ranging from potatoes, cereals, and biscuits, to coffee and meat products, after heat treatment [1,2]. Studies have shown that AA is toxic in a multitude of ways, which include neurotoxic, hepatotoxic, androgenic, developmentally toxic, and carcinogenic. It is classified as a group 2A substance by the International Agency for Research on Cancer (IARC), and its toxic effects have been investigated intensively [3,4]. A study on dietary exposure to acrylamide in the Chinese population showed that the mean dietary exposure to acrylamide was 0.1531 µg/kg body weight/day for men and 0.1554 µg/kg body weight/day for women [5]. The maximum limit of AA in drinking water is 1.00 μg/L as stipulated by the WHO, while China stipulates that it cannot be higher than 0.50 μg/L and many other countries stipulate a maximum limit of 0.25 mg/L [6,7].

Ochratoxin (OT) is a secondary metabolite secreted by *Aspergillus ochraceus*, *Penicillium verrucosum*, and other *Penicillium* species, discovered in South Africa in 1965, and includes seven structural analogs such as OTA, OTB, and OTC, of which OTA is the most widely distributed toxin [8]. OTA can be found in maize, wheat, barley, flour, coffee, and mixed feed, among other agricultural goods [9]. OTA is difficult to remove from food under typical cooking settings, even when treated at 250 °C for a few minutes, since it is very stable in acidic environments and exceedingly thermostable; therefore, OTA may be found in cereal goods, beer, and roasted coffee [10,11]. Iqbal et al. [12] found that 41% of 115 chicken and 80 egg samples (mean OTA of 1.41 ± 0.70 μg/kg) and 35% of egg samples (mean OTA of 1.17 ± 0.42 μg/kg) were contaminated with OTA. OTA has been reported to cause immunotoxicity, genotoxicity, and neurotoxicity, and is classified as a class 2B substance [13,14]. Based on the nephrotoxicity of OTA, the Joint Food and Agricultural Organization (FAO)/World Health Organization (WHO) Expert Committee on Food Additives (JECFA) recommends a Provisional Tolerable Daily Intake (PTDI) of 14 ng/kg/body weight for OTA [7].

The most common way of absorbing AA and OTA is oral ingestion through food intake [15,16,17]. Some researchers have confirmed that AA and OTA have intestinal toxicity [18,19]. Rita Pernice et al. [20] used Caco-2 cells as a model and found that AA rapidly entered small intestinal cells, significantly reduced intracellular GSH levels, and promoted the generation of large amounts of ROS free radicals, causing oxidative stress, which inhibited normal cell function and led to cell death or apoptosis. Similarly, Romero et al. [21], using a Caco-2 cell model, discovered that OTA significantly declined TEER values and mRNA expression levels of TJ proteins.

Compared to single mycotoxins, simultaneous ingestion of multiple mycotoxins from food may lead to different types of interactions, such as additive and synergistic, or antagonistic effects [22]. For example, Bensassi et al. [23] examined the survival, cell cycle, and mitochondrial transmembrane potential of human colon cancer cells (HCT116) exposed to deoxynivalenol (DON) and zearalenone (ZEA) and showed that the effect of the combination of DON and ZEA on these indicators was lower than the effect of the toxins alone, showing an antagonistic effect. Otherwise, due to the presence of OTA in various foods, numerous studies have been carried out to develop methods to mitigate OTA-induced toxicity [24,25]. Foods are heat-treated to reduce OTA, but the possibility of acrylamide production during this process cannot be ruled out [26]. Some studies have confirmed that AA and OTA can both be found in the same foods, including cereals, coffee, and beer [17,26,27]. In general, OTA growth in corn during storage can be controlled through controlled atmospheres, preservatives, or natural inhibitors [28], and different processes optimized during corn processing, such as infrared heating, extrusion cooking, or microwave heating, can also help reduce AA in corn-based products [29].

Therefore, research into the combined toxic effects of AA and OTA is necessary; however, no studies on the combined effects of AA and OTA on Caco-2 cells have been published. Thus, the goal of this study was to investigate the toxicity and potential mechanism of the AA and OTA interaction in Caco-2 cells. Our results may provide new insights into AA- and OTA-induced epithelial tight junction dysfunction.

## 2. Materials and Methods

### 2.1. Materials

AA was purchased from Aladdin Chemical Co., Ltd. (Shanghai, China). OTA was purchased from Pribolab Biological Engineering Co., Ltd. (Qingdao, China). The Cell Counting Kit-8 (CCK-8) kit was bought from APExBIO Technology (Houston, TX, USA). Lactate dehydrogenase (LDH), Bicinconic Acid Assay (BCA), and reactive oxygen species (ROS) kits were obtained from Shanghai Beyotime Biotechnology Co., Ltd. (Shanghai, China). Lucifer yellow (LY) was purchased from Sigma-Aldrich (St. Louis, MO, USA). ELISA kits were sourced from Boster Company (Wuhan, China). Antibodies against TJ (occludin, claudin-1, and ZO-1) and corresponding secondary antibodies were purchased from Abcam (Shanghai, China). All other chemical reagents in this research were of analytical grade.

### 2.2. Cell Culture

The Caco-2 cell line was purchased from the Cell Bank of the Chinese Academy of Sciences (Shanghai, China) and cultured in a Dulbecco’s modified eagle medium (DMEM) complete medium supplemented with 16% (*v*/*v*) fetal bovine serum (FBS), antibiotics (100 U/mL penicillin and 100 μg/mL streptomycin), 1% non-essential amino acid (NEAA) at 37 °C, and 5% CO_2_ in a humidified incubator. When the cells reached 80% to 90% confluence, they were detached for 4 min with 0.25% trypsin-EDTA, and then re-inoculated at a 1:3 dilution into a new T75 flask. The cells were used between 15 and 35 generations for the experiments.

### 2.3. Cell Viability

The proliferative toxicity of AA and OTA was measured quantitatively using the CCK-8 kit, as directed by the manufacturer. In short, Caco-2 cells were seeded at 1 × 10^5^ cells/mL in 96-well plates using 100 μL of complete proliferation medium. Different concentrations of poisons were exposed individually or collectively for 24 h or 48 h after 24 h of culture. After that, 100 μL of the CCK-8 solution was added to each well and kept incubating for 2 h. A microplate reader was employed to measure the absorbance at 450 nm.

### 2.4. LDH Assay

The leakage of LDH indicates irreversible cell death due to disruption of the cell membrane [30]. After Caco-2 cells had been tested with AA and OTA for 24 h, the cell supernatant was collected and tested with an LDH assay kit based on the instructions provided by the manufacturer.

### 2.5. Transepithelial Electrical Resistance (TEER) Assay

Recording TEER was performed with a Millicell-ERS instrument (Millipore, Bedford, MA, USA) to assess the impact of AA and OTA on cell membrane integrity. Caco-2 cells were seeded into the apical lumen of 6- or 12-well Transwell plates (Corning, New York, NY, USA) at the density of 2 × 10^5^ cells/cm^2^. For 21 days, the DMEM medium was replaced every other day. The TEER values were based on the following formula: TEER (Ω.cm^2^) = [Measured value (Ω) − Background value (Ω) × A (cm^2^)], where the background is cell-free inserts and A is the area of the membrane. The TEER value of cells was higher than 300 Ω·cm^2^ and was used for further experiments. Experiments were repeated three times with five replicates for each treatment.

### 2.6. Paracellular Flux Assay

LY is a common paracellular tracer that reflects the extent of cell membrane disruption [31]. Caco-2 cells were cultured for 21 days, then AA and OTA were added individually or collectively to Transwell chambers for 24 h. LY at a concentration of 100 μg/mL was added to the chambers for 2 h, and 200 μL of samples were collected from the basolateral chambers. Fluorescence intensity was determined at 410 nm and 520 nm excitation and emission wavelengths using a microplate reader.

### 2.7. Detection of Production Level of ROS

Caco-2 cells were cultured in 6-well plates and exposed to AA and OTA individually or jointly for 24 h, then the medium was removed and washed with PBS twice. The cells were collected and incubated with 2,7-Dichlorodihydrofluorescein diacetate (DCFH-DA) diluted in FBS-free medium for 30 min at 37 °C in the dark. The production of ROS was measured with flow cytometry.

### 2.8. ELISA Assay

After 24 h of incubation in the presence of AA and OTA, the cells were centrifuged at 1000 r/min for 5 min, and the cell culture supernatant was collected for ELISA assay. The ELISA kits were employed to determine the contents of TNF-α, IL-6, IL-1β, and IL-10 in the supernatant of the Caco-2 cells following the manufacturer’s instructions.

### 2.9. Western Blot Analysis

After AA and OTA were applied individually or jointly to Caco-2 cells for 24 h, Western blot experiments were performed, referring to the method of Jiang et al. [32], and corresponding improvements were made according to the actual situation. Cells were collected by washing two times with Phosphate-buffered saline (PBS), then the total protein was extracted by RIPA buffer, and protein concentration was assayed with an enhanced BCA protein detection kit. After quantification, proteins were applied to 12% Sodium dodecyl sulfate polyacrylamide gel electrophoresis (SDS-PAGE) and transferred onto Polyvinylidene Fluoride (PVDF) membranes. After PVDF membranes were blocked with 5% BSA in room-temperature conditions for 1 h, they were incubated overnight at 4 °C with primary antibodies, including anti-claudin-1 (1:1000, abcam, Shanghai, China), anti-ZO-1 (1:1000, abcam, Shanghai, China), anti-occludin (1:1000, abcam, Shanghai, China), and anti-β-actin (1:1000, abcam, Shanghai, China). After three washes with TBST, the membranes were incubated at room temperature for 1 h with the secondary antibody (1:10,000, ZSGB Biotechnology, Beijing, China). Afterward, visualization of the protein bands was carried out with an enhanced chemiluminescence (ECL) detection reagent. Signal plots were quantified with the Image J software (version 2.1.0, National Institutes of Health, Bethesda, MD, USA, 2006) and the target protein expression levels were normalized against the intensity of β-actin and expressed as a percentage of the blank group.

### 2.10. Interaction and Correlation Analysis

The comparison of measured and expected theoretical values can be used to evaluate the interaction of toxins [33,34]. The expected values were calculated by adding the mean value after exposure to a substance alone to the mean value after exposure to another toxicant [31].

Significance analysis was performed between expected and measured values through an unpaired *t*-test. *p* < 0.05 was considered significant.

In order to analyze the interaction types of AA and OTA, the expected values of cell viability, TEER value, LY permeability, LDH release, inflammatory factor expression level, and TJ proteins expression were calculated, respectively. The definitions of the different effects are explained below:

The additive effects were defined as the non-significant difference between expected and measured values (*p* > 0.05).

The synergistic effects were defined as the expected values of cell viability, TEER value, anti-inflammatory factor level, and TJ proteins expression being significantly higher than the measured values of these indicators, while LY permeability, the expected values of ROS generation, LDH release, and pro-inflammatory factor levels were significantly lower than the measured values of these indicators.

The antagonistic effects were defined as the expected values of cell viability, TEER value, anti-inflammatory factor level, and TJ proteins expression being significantly lower than the measured values of these indicators, while LY permeability, the expected values of ROS generation, LDH release, and pro-inflammatory factor levels were significantly higher than the measured values of these indicators.

Correlations among cell viability, TEER value, LY permeability, LDH release, inflammatory factor expression level, and expression of TJ proteins in Caco-2 cells exposed to individual and combined AA and OTA were evaluated with Spearman’s correlations.

### 2.11. Statistical Analysis

These data were analyzed using SPSS 26.0 (IBM, Chicago, IL, USA), and the results were expressed as mean standard deviation (Mean ± SD). One-way analysis of variance (ANOVA) was used to analyze differences between groups. GraphPad Prism 8.0.2 software (GraphPad Inc., San Diego, CA, USA) was used for drawing. Two-tailed *p* < 0.05 was considered to indicate a statistically significant difference.

## 3. Results and Discussion

### 3.1. Cytotoxicity

In this experiment, the cytotoxicity of intestinal epithelial Caco-2 cells was assessed using different concentrations of AA and OTA, and the results of the CCK-8 assay are shown in Figure 1. From Figure 1a,b, it can be seen that, compared to the normal control group, the survival rate of Caco-2 cells was significantly decreased after AA or OTA exposure with different times (24 or 48 h) (*p* < 0.05). The cell viability significantly dropped to 12.54% and 16.17% of the control when the concentration of AA and OTA was increased to 40 mM and 40 μM for 24 h, respectively. In addition, the values after 48 h were clearly lower than those after 24 h (*p* < 0.05). The toxicity of AA or OTA increased with increasing concentration, showing a concentration-dependent effect. Furthermore, the IC_50_ values of AA and OTA were found to be 5 mM and 5 μM, respectively.

According to the results of cell viability individually, three different concentrations around the IC_50_ of AA (2.5, 5, and 10 mM) and OTA (2.5, 5, and 10 μM) were subsequently selected to evaluate their joint toxicity. As shown in Figure 1c, the cell viability decreased significantly (from 68% to 30%) in the simultaneous presence of AA and OTA (*p* < 0.05).

AA and OTA are commonly found in food, and the co-occurrence of AA and OTA in food may create a serious risk to human health. Nevertheless, most studies have concentrated on the toxic effects of individual toxins. Recently, Min Cheol Pyo et al. [35] studied the toxicity of OTA and AA in combination on human kidney and liver cells, and showed that the synergistic toxicity of OTA and AA may lead to both nephrotoxicity and hepatotoxicity. According to previous reports, as small molecular compounds, AA and OTA are highly absorbed through the gastrointestinal system, thus the intestine is the primary target organ of AA and OTA toxicology. However, until now, no studies on the combined intestinal toxicity of AA and OTA have been reported.

After inoculation of Caco-2 cells onto permeable membranes for approximately 21 days, Caco-2 cells can autonomously differentiate into monolayers that express many of the typical characteristics of absorptive intestinal cells, such as brush border layers, tight junction, and having various transport systems and metabolic enzymes, which can imitate the intestinal barrier in vivo [36,37,38]. Therefore, the Caco-2 cell line is one of the most frequent and commonly employed in vitro models for studying the passage of fungal toxins through the intestinal membrane, enterocytes, or intestinal absorption [39,40].

### 3.2. The LDH Leakage of Caco-2 Cells

The LDH release in different treatment groups is illustrated in Figure 2. Compared to the normal group, LDH content in the supernatant of AA or OTA treatment groups at different concentrations was remarkably increased (*p* < 0.05), among which the LDH release in the 10 mM AA treatment group was remarkably higher than that in the low-concentration AA group (*p* < 0.05), with a dose-effect relationship, while in the OTA-alone treatment group, no remarkable changes in LDH leakage were observed (*p* > 0.05). The combined effect of the AA + OTA group on LDH release was remarkably greater than that of the individual treatment groups (*p* < 0.05), indicating that the effect of high AA + OTA combination treatment on Caco-2 cells was greater than that caused by AA or OTA individually.

### 3.3. The TEER Values of Caco-2 Cells

The elevation and decline of TEER values are directly correlated with the degree of tight junction (TJ) integrity and, therefore, changes in TEER values are commonly used to reflect the integrity of the cellular TJ. As Figure 3 shows, TEER values in differentiated Caco-2 cells were dramatically reduced (*p* < 0.05) by AA and OTA individually or collectively after 24 h, among which AA (10 mM) and OTA (10 μM) individual groups had significantly lower TEER values compared to the other individual dose groups (*p* < 0.05). In addition, the TEER values of the AA + OTA groups were notably below those of the AA or OTA groups (*p* < 0.05).

### 3.4. The Permeability of Caco-2 Cells

As shown in Figure 4, the LY permeabilization was increased (*p* < 0.05) when Caco-2 cells were treated with low concentrations of AA and OTA individually or jointly, compared to the control group. Among them, the groups with low concentrations of AA (2.5 and 5 mM), OTA (2.5 and 5 μM), and AA + OTA (2.5 mM + 2.5 μM and 5 mM + 5 μM) were not significantly different (*p* > 0.05). In the AA + OTA group, the LY permeabilization of the 10 mM + 10 μM concentration group was remarkably higher than that of the single treatment ones (*p* < 0.05).

The activity of the intestinal epithelium and the integrity of the intestinal barrier were evaluated by cell viability, TEER values, LY permeability, and LDH release. It was found that viability and TEER values were significantly decreased after AA- and OTA-induced Caco-2 cells in our research. Increased paracellular transport, apoptosis, or transcellular permeability can lead to epithelial permeability [41]. As previously reported [31,42], studies found that differentiated cells exposed to a mixture of AFM1 and OAT resulted in decreased TEER values, enhanced LY permeability, and disruption of intestinal barrier function. These results indicated that AA- and OTA-induced reduction in cell viability may be important factors in the altered permeability of enterocytes.

### 3.5. The ROS Level of Caco-2 Cells

To further explore the influences of AA and OTA on the oxidative stress of Caco-2 cells, the ROS generation was measured with flow cytometry and is shown in Figure 5. Compared to the control group, after being treated with different concentrations of AA or OTA, ROS content was remarkably increased (*p* < 0.05), indicating that AA and OTA can cause Caco-2 cells to generate a large amount of ROS and induce oxidative stress. The AA + OTA combination treatment group generated more ROS in Caco-2 cells than the AA or OTA groups treated separately, showing that joint toxicity of the two toxins occurred.

### 3.6. The Secretion of Inflammatory Cytokines of Caco-2 Cells

From Figure 6, compared to the control group, IL-1β content in all groups was markedly increased (*p* < 0.05), except for the AA treatment group at the low and medium concentrations (2.5 mM and 5 mM), and the AA + OTA treatment group at the low concentration (5 mM + 5 μM). Compared with the control group, individual or joint treatment with a high concentration of AA and OTA significantly increased the content of IL-6 in the supernatant of Caco-2 cells (*p* < 0.05). The TNF-α levels in the cell supernatants in all groups were considerably increased (*p* < 0.05), except for the low-concentration OTA (2.5 μM) group. As shown in Figure 6d, compared to the control group, all groups significantly reduced the content of IL-10 in Caco-2 cells (*p* < 0.05). Our study reported that AA- and OTA-induced Caco-2 cells increased the amounts of IL-1β, IL-6, and TNF-α, while decreasing the IL-10 level. The above results suggested that AA and OTA could cause inflammatory damage to cells by increasing pro-inflammatory factor levels as well as reducing anti-inflammatory factor levels in Caco-2 cells.

### 3.7. The TJ Protein Expression of Caco-2 Cells

To explore the potential mechanism of AA- and OTA-induced increase in the permeability of Caco-2 cell monolayers, this study used Western blot to quantify TJ protein expression (Figure 7). After exposure to AA, the claudin-1 expression was observed to be negatively correlated with AA concentration. When the AA concentration increased to 10 mM, the value was decreased by approximately 65.55%. Likewise, the addition of OTA also led to a slight decrease in the claudin-1 protein expression. For occludin and ZO-1 protein expressions (Figure 7c), OTA treatment was prone to increase the expression levels, while AA treatment with or without the presence of OTA was more likely to decrease the expression levels. Meanwhile, the expression of all the above-mentioned proteins in cells under the co-existence of AA and OTA was much lower than that with individual exposure to AA or OTA (*p* < 0.05). A significant joint effect was observed when high concentrations of combined AA and OTA were used.

The intestinal barrier includes surface mucus, the epithelial layer, and immune defenses [43,44]. The TJ of polarized epithelial cells modulates the barrier function of the mucosal surface. In addition, structural and functional proteins of TJ include claudins and occludin, as well as ZO-1 [45]. There are few studies on the effect of AA and OTA on the integrity of the intestinal barrier that we are aware of. Research reported that exposure to AA reduced the expression of TJ proteins in the Caco-2 intestinal cell line model [46]. Alizadeh et al. [47] showed that OTA exposure down-regulated TJ protein expression levels in Caco-2 cells. In the research of McLaughlin et al. [48], the expressions of claudin-3 and claudin-4 were decreased in Caco-2 cells after 24 h of OTA exposure. Our Western blotting analysis showed that AA and OTA could inhibit the expression of TJ proteins. Barrier dysfunction in the gut causes invasion of intestinal microorganisms, inducing over-immunization of the body’s immune cells, which leads to intestinal inflammation [41].

### 3.8. The Interactive Effects of the Combined Treatment of AA and OTA

Synergistic effects of AA and OTA exposure were found in the Caco-2 cells’ viability (Figure 8a). Results for the TEER values showed that the interactive effects changed from additive effects to synergistic effects as the concentration of AA and OTA increased from low dose (2.5 mM + 2.5 μM) to high doses (5 mM + 5 μM and 10 mM + 10 μM) (Figure 8b). For the paracellular flux of LY, ROS production, and LDH release, there were additive effects where the difference between the measured and predicted values was not significant (*p* > 0.05) (Figure 8c–e). For the inflammatory cytokines, the antagonistic effects of AA and OTA were evident on the level of IL-10 and IL-1β, with the measured values lower than the expected ones (*p* < 0.05) (Figure 8f,i). Synergistic effects of IL-6 and TNF-α levels were noted following exposure of cells to AA and OTA mixtures (Figure 8g,h). For claudin-1 expression, a synergistic effect was obtained at low and medium doses of mixture, while at high concentrations of AA + OTA (10 mM + 10 μM), an antagonistic effect was observed (Figure 8j). Synergistic effects were found in the expression of occludin and ZO-1 proteins at three concentrations of AA and OTA, with the measured values significantly lower than expected ones (*p* < 0.05) (Figure 8k,l). Thus, it can be concluded that, for most of the detection indicators, AA and OTA combined exposure tends to produce synergistic toxicity in Caco-2 cells.

### 3.9. Correlation Analysis

Figure 9 shows the correlations between cell viability, TEER value, LY permeability, LDH release, inflammatory factor expression level, and TJ protein expression. There is a significant positive correlation (*p* < 0.05) between cell viability and TEER values in Caco-2 cell monolayers. A significantly negative (*p* < 0.05) correlation is observed between the TEER values and LY permeability. Furthermore, the results show a positive correlation (*p* < 0.05) between TEER values and TJ protein (claudin-1, occludin, and ZO-1) expression, indicating that the increased epithelial permeability is related to the disruption of TJ integrity.

In this study, combined AA and OTA showed different types of interactions in disrupting the intestinal barrier, including additive, synergistic, and antagonistic effects. Generally, the co-existence of other compounds with a common pattern of action and/or the same cellular target can result in synergistic or additive interactions [49]. It has been shown that exposure to AA and OTA leads to oxidative DNA damage, which is the main mechanism of cytotoxicity [50,51]. This may explain the synergistic and additive interaction effects that we observed in this research. Moreover, the antagonistic effect of AA and OTA is probably accounted for through intracellular competition for glutathione (GSH). Since electrophiles generated from the metabolism of OTA with hydroquinone–quinine reduce GSH to produce GSH conjugates, AA can also spontaneously or enzymatically conjugate with GSH to form its corresponding GSH conjugates [52,53,54]. In practice, the type of interaction between multiple toxins depends on the concentration of toxin used, the duration of exposure, the type of test model chosen, and the indicators assessed [55]. Therefore, the potential mechanisms of AA- and OTA-induced intestinal barrier dysfunction must yet be explored.

## 4. Conclusions

In conclusion, in the present research, we demonstrated that AA and OTA disrupt the intestinal epithelial barrier by reducing TEER values, increasing the LY permeability, LDH release, and ROS production, improving pro-inflammatory factor amounts, reducing anti-inflammatory factor levels, and inhibiting the expression of TJ proteins. In addition, the collective effects of AA and OTA showed greater toxicity than single compounds and, for most measures of intestinal barrier function, combined AA and OTA exposure tended to produce synergistic toxicity on Caco-2 cells. Furthermore, to our knowledge, this is the first study on the combined effects of AA and OTA on Caco-2 cells. These findings provide data on the injury to the intestinal tract caused by AA and OTA, and may provide scientific guidance and an experimental basis for dietary nutrition. People can reduce the AA and OTA content in food, or reduce their production, by adjusting the temperature (below 110 °C is a relatively safe temperature), time (frying time is limited to 3 min), and moisture content (taking pre-drying treatment to reduce the moisture content of raw materials) during cooking or processing in life [6].

## Figures and Tables

**Figure 1 foods-12-01318-f001:**
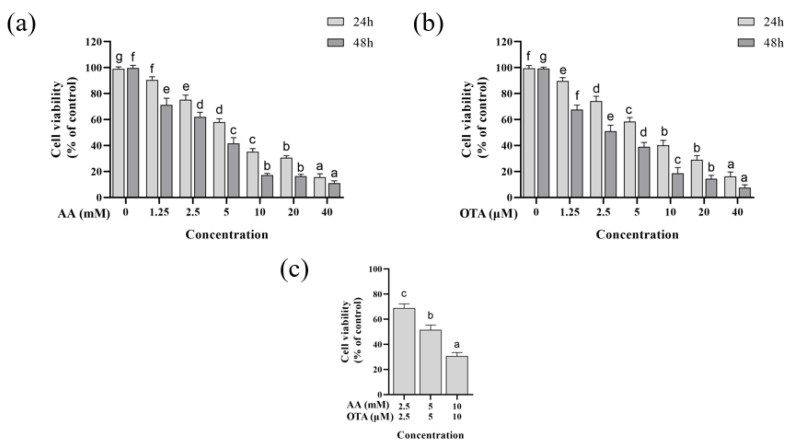
Effect of AA and OTA at different concentrations on the cell viability in Caco-2 cells. (**a**) Cell viability after 24 h or 48 h treatment with AA (0–40 mM). (**b**) Cell viability after 24 h or 48 h treatment with OTA (0–40 μM). (**c**) Cell viability of Caco-2 cells exposed to mixtures of AA and OTA for 24 h. ^a–g^ Different letters indicate significant differences, *p* < 0.05.

**Figure 2 foods-12-01318-f002:**
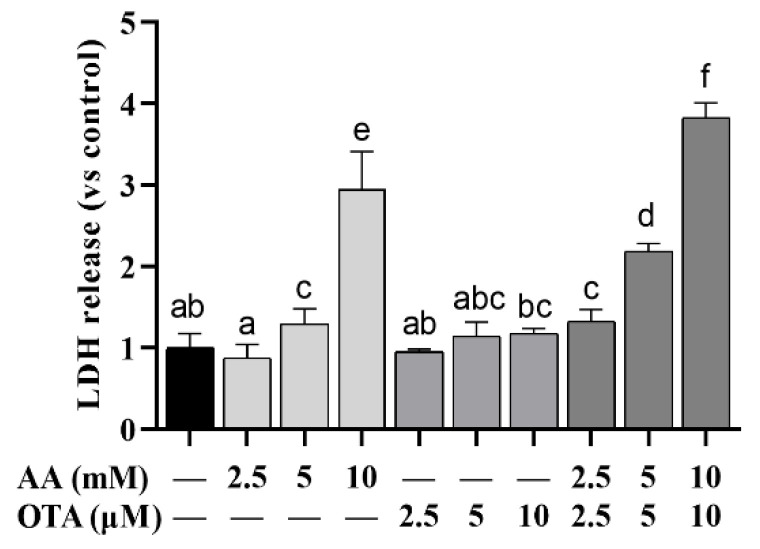
Effect of AA and OTA on LDH release. ^a–f^ Different letters indicate significant differences, *p* < 0.05.

**Figure 3 foods-12-01318-f003:**
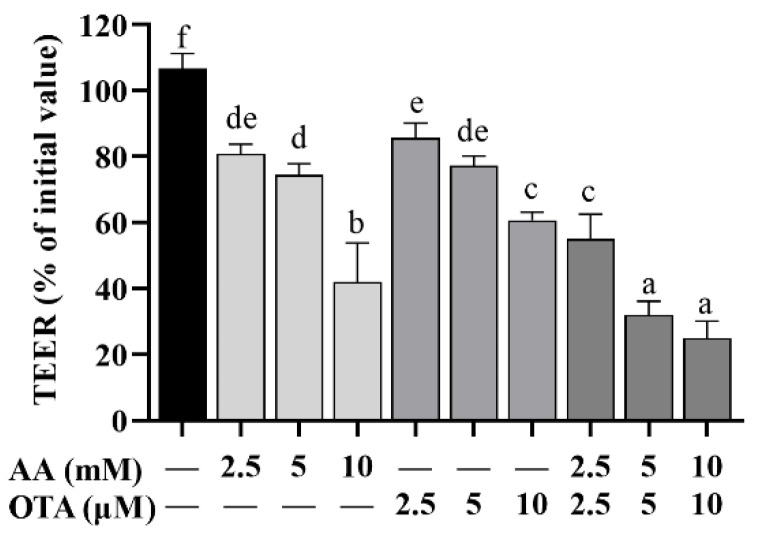
Effect of AA and OTA on TEER values. ^a–f^ Different letters indicate significant differences, *p* < 0.05.

**Figure 4 foods-12-01318-f004:**
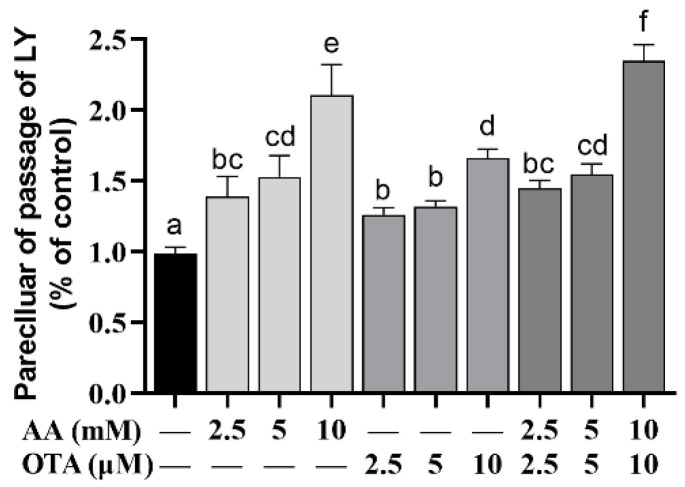
Effect of AA and OTA on the permeability of LY. ^a–f^ Different letters indicate significant differences, *p* < 0.05.

**Figure 5 foods-12-01318-f005:**
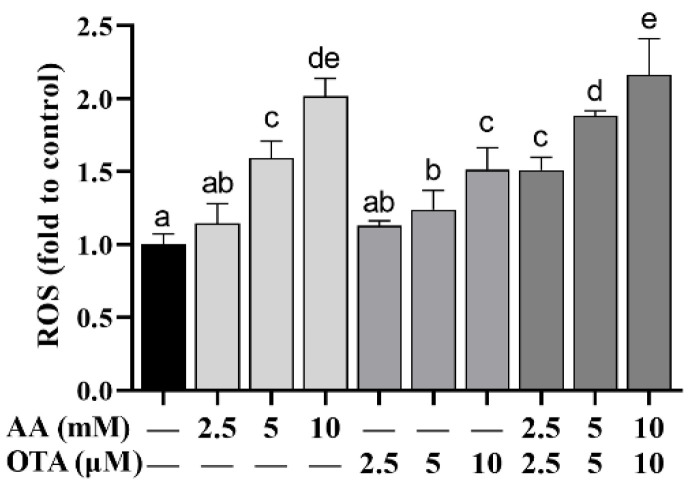
Effect of AA and OTA on ROS production. ^a–e^ Different letters indicate significant differences, *p* < 0.05.

**Figure 6 foods-12-01318-f006:**
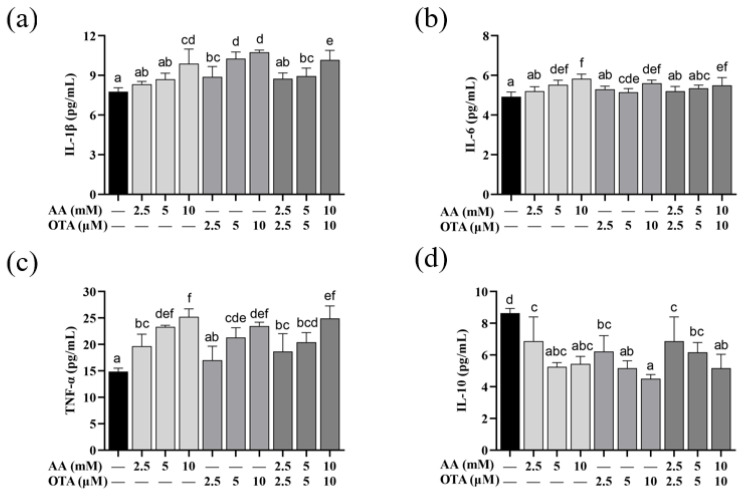
Effect of AA and OTA on inflammatory factors in the Caco-2 cells after 24 h. (**a**) IL-1β content. (**b**) IL-6 content. (**c**) TNF-α content. (**d**) IL-10 content. ^a–f^ Letters indicate differences between groups, *p* < 0.05.

**Figure 7 foods-12-01318-f007:**
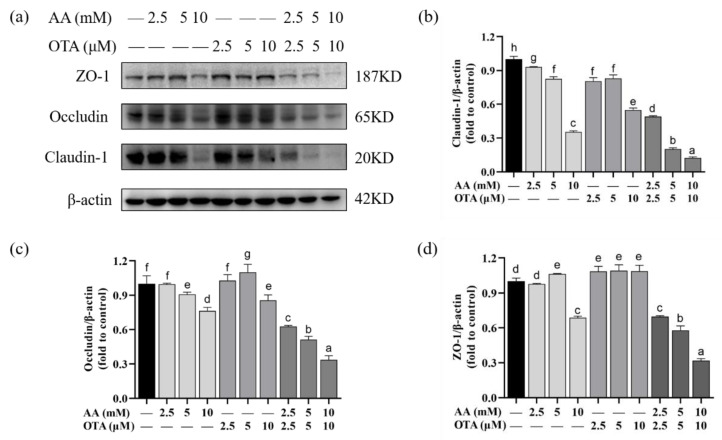
Effect of AA and OTA on the expression of TJ proteins in the Caco-2 cells. (**a**) TJ proteins were determined by SDS-PAGE. (**b**–**d**) Relative protein expression of TJ from different groups. Values are shown as means ± SEM (*n* = 3). ^a–h^ Letters indicate differences between groups, *p* < 0.05.

**Figure 8 foods-12-01318-f008:**
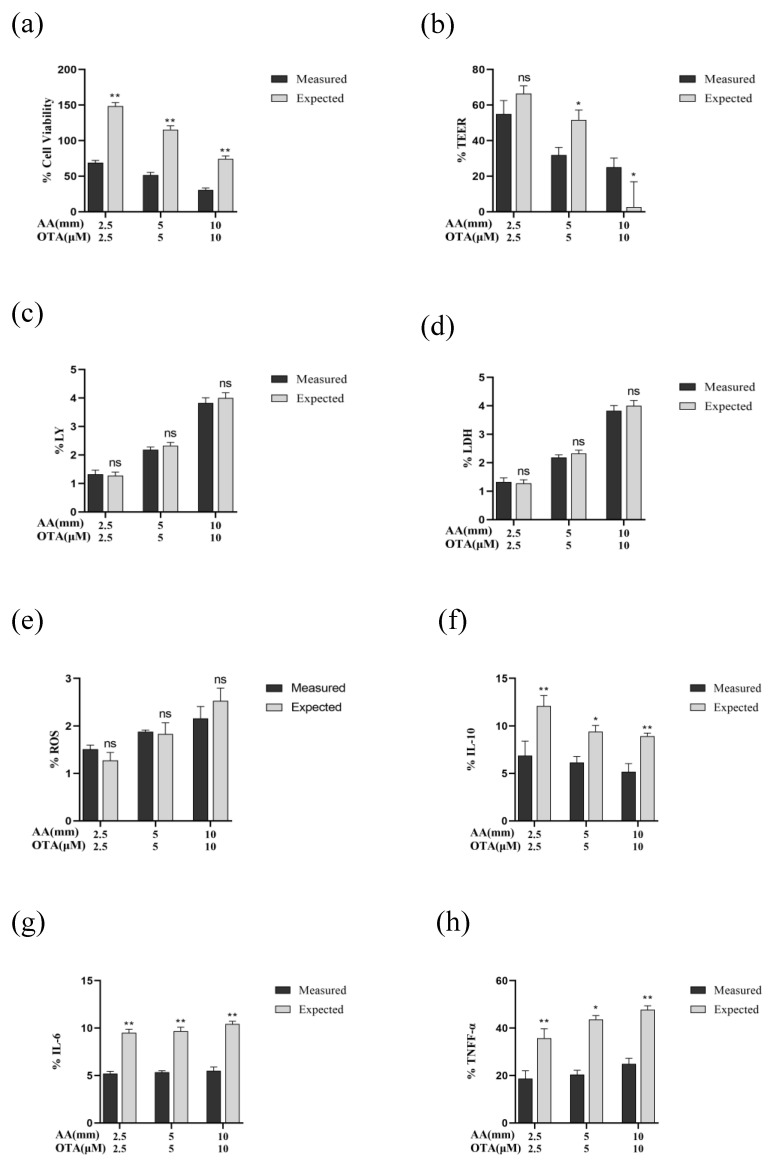
Interactive cytotoxic effects of AA and OTA collectively in the Caco-2 cells after 24 h (**a**–**l**). Data are presented as a percentage of unprocessed controls for each parameter. (*) *p* < 0.05; (**) *p* < 0.01. And ns means no significant difference.

**Figure 9 foods-12-01318-f009:**
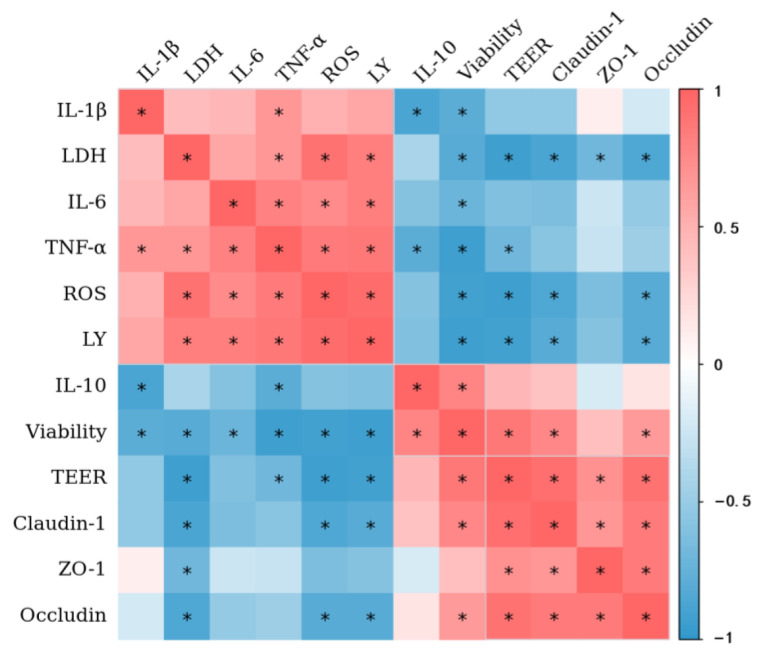
The heat map shows a correlation between cell viability, LDH release, TEER, the paracellular flux of LY, ROS production, inflammatory cytokines, and TJ proteins of Caco-2 cells. The heat map represents a visual indication of the correlation values among each pair of parameters represented in the respective rows and columns of the matrix. (*) *p* < 0.05.

## Data Availability

Data is contained within the article.

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
