# Peer review of "Combined Effects of Acrylamide and Ochratoxin A on the Intestinal Barrier in Caco-2 Cells"

_foods, 2023, doi:10.3390/foods12061318_

Round 1
Reviewer 1 Report
The article "Combinational Effects of Acrylamide and Ochratoxin A on Intestinal Barrier in Caco-2 Cells" is well written and informative regarding the effects of AA and OTA collectively on Caco-2 cells and authors demonstrated that AA and OTA disrupt the intestinal epithelial barrier through reducing TEER values, increasing the LY permeability, LDH release, and ROS production, improving pro-inflammatory factors amounts and reducing anti-inflammatory factor level, inhibiting the expression of TJ proteins. However, I could not see the novelty in this paper. If the effect of individual constituents is known, then it is expected that both the constituents will cause more toxicity on Caco2 cells. To make it more informative and meaningful, the available data on toxic doses and admissible doses need to be discussed in Introduction and then need to be discussed along with the results obtained in the present work. Authors may resubmit after addressing these major points and other comments as given below:
Other comments:
Introductory one sentence to be included in the abstract.
First time full words and then abbreviations in the Abstract and Text.
Line 38 class B …mention category.
93 correct to replicates
Section 2.8: Provide brief description
Section 2.10: no need to define terms
Section 2.11: There was significant ….. should come in results or change
Line 164: correct spellings of toxicity
Figure 1: Of control or with respect to control. It is better to provide combined graph to depict the comparative values of different treatments.
Figures: Provide the assay as given in Figure 7…as determined by…
Author Response
Response to reviewer
Reviewer #1:
The article "Combinational Effects of Acrylamide and Ochratoxin A on Intestinal Barrier in Caco-2 Cells" is well written and informative regarding the effects of AA and OTA collectively on Caco-2 cells and authors demonstrated that AA and OTA disrupt the intestinal epithelial barrier through reducing TEER values, increasing the LY permeability, LDH release, and ROS production, improving pro-inflammatory factors amounts and reducing anti-inflammatory factor level, inhibiting the expression of TJ proteins. However, I could not see the novelty in this paper. If the effect of individual constituents is known, then it is expected that both the constituents will cause more toxicity on Caco2 cells. To make it more informative and meaningful, the available data on toxic doses and admissible doses need to be discussed in Introduction and then need to be discussed along with the results obtained in the present work. Authors may resubmit after addressing these major points and other comments as given below:
Response: We appreciate your positive comments and advice. According to your comments, the manuscript has been carefully revised. We hope that you could be satisfied with our corrections and changes.
Compared to single mycotoxins, simultaneous ingestion of multiple mycotoxins in food may lead to different types of interactions, such as additive and synergistic or antagonistic effects [1]. For example, Bensassi et al. examined the survival, cell cycle and mitochondrial transmembrane potential of human colon cancer cells (HCT116) exposed to DON and ZEA and showed that the effect of the combination of DON and ZEA on these indicators was lower than the effect of the toxin alone, showing an antagonistic effect [2].
And the available data on toxic doses and admissible doses has been added in the revised manuscript. Please find Introduction part.
[1] Ficheux A S, Sibiril Y, Parent-Massin D. Co-exposure of Fusarium mycotoxins: in vitro myelotoxicity assessment on human hematopoietic progenitors[J]. Toxicon, 2012, 60(6): 1171-1179.
[2] Bensassi F, Gallerne C, Hajlaoui M R, et al. In vitro investigation of toxicological interactions between the fusariotoxins deoxynivalenol and zearalenone[J]. Toxicon, 2014, 84: 1-6.
Other comments:
Introductory one sentence to be included in the abstract.
Response: Thank you very much for your careful review. The introductory have been supplemented in the revised manuscript.
First time full words and then abbreviations in the Abstract and Text.
Response: Thank you for your significant reminder. The full description of words and abbreviations have been changed in the revised manuscript.
Line 38 class B …mention category.
93 correct to replicates
Response: Thank you for your significant reminder. According to your advice, in the revised manuscript we have corrected these errors.
Section 2.8: Provide brief description
Response: We gratefully appreciate for your comment. According to your suggestion, the brief description of ELISA assay was added in the revised manuscript. Please see section 2.8.
Section 2.10: no need to define terms
Response: Thank you for your significant reminder. But we think it is necessary to explain the terms because that different evaluation criteria were applied for different measurement indicators.
Section 2.11: There was significant ….. should come in results or change
Response: Thank you for your valuable suggestion. According to your suggestion, this part have been rewritten in the revised manuscript. Please see section 2.11.
Line 164: correct spellings of toxicity
Response: We gratefully appreciate for your comment. According to your advice, in the revised manuscript we have corrected this error.
Figure 1: Of control or with respect to control. It is better to provide combined graph to depict the comparative values of different treatments.
Response: Thank you for your significant reminder. We have redrawn the picture based on your suggestions in the revised manuscript. Please refer to Figure 1.
Figures: Provide the assay as given in Figure 7…as determined by…
Response: We gratefully appreciate for your comment. Corresponding revision has been made in the revised manuscript. Please find Section 2.9 and Figure 7.

Reviewer 2 Report
This paper is an example of a very well conducted study with a wide range of pertinent methods that are properly described and carried out with expertise. The major problem of this study lies in why were the combinations of Ochratoxin-A and acrylamide decided on in the first place? The Ochr-A is indeed heat stable but would toxicity bioactivity assessment be compromised at the heat temperatures needed for Acrylamide synthesis. Moreover, are the cereal choices used in this study all equally susceptible to acrylamide generation. These are basic questions that should be diescussed before a rational of combining the two toxins is made.
The use of IC50 values to decide on the range of Ochr-A and acrylamide is done correctly; however, under real conditions could they actually be used for a range of toxins that are identical (e.g. 2.3-10um). It is interesting that the combination of the two toxins produced different types of interactions dependent on bioactivity measured, some being additive, synergistic or antagonistic... can the authors explain why these interactions would have different threshold effects.
Finally, the final message that these results can lead to nutritional decisions needs to be expanded on. What are the decisions that are suggested.
Author Response
Response to reviewer
Reviewer #2:
This paper is an example of a very well conducted study with a wide range of pertinent methods that are properly described and carried out with expertise. The major problem of this study lies in why were the combinations of Ochratoxin-A and acrylamide decided on in the first place? The Ochr-A is indeed heat stable but would toxicity bioactivity assessment be compromised at the heat temperatures needed for Acrylamide synthesis. Moreover, are the cereal choices used in this study all equally susceptible to acrylamide generation. These are basic questions that should be discussed before a rational of combining the two toxins is made.
Response: Thank you very much for your great support for the acceptance of our manuscript in Foods. We feel very lucky that our manuscript went to you as the valuable comments from you helped us with the improvement of our manuscript. According to your guideline, the manuscript has been carefully revised.
Because of the presence of OTA in various foods, many studies have been conducted to develop methods to mitigate OTA-induced toxicity. Foods are heat-treated to reduce OTA, but the possibility of acrylamide production during this process cannot be ruled out [1]. It has been confirmed in several studies that AA and OTA can be present together in the same food, including cereals, coffee and beer [1-3]. The combined effects of AA and OTA on intestinal barrier damage in in vivo or in vitro models have not been reported, and given the toxic hazard of both toxins and their common co-occurrence in heat processed foods, it is essential to assess the toxic damage to the intestinal barrier from their combined effects.
[1] Alkhalifah, D. H.; EL-Sideek, L. E.; Deabes, M. M.; Elgammal, M. H.; Farag Zaied, S. A., Comparing effect of Egyptian, Saudi Arabian coffee cup preparations on Ochratoxin A and Acrylamide content. International Journal of Academic Research 2013, 5 (3), 168-177.
[2] Bogdanova, E.; Rozentale, I.; Pugajeva, I.; Emecheta, E. E.; Bartkevics, V., Occurrence and risk assessment of mycotoxins, acrylamide, and furan in Latvian beer. Food Additives & Contaminants: Part B 2018, 11 (2), 126-137.
[3] Seal, C. J.; de Mul, A.; Eisenbrand, G.; Haverkort, A.; Franke, K.; Lalljie, S.; Mykkänen, H.; Reimerdes, E.; Scholz, G.; Somoza, V., Risk-benefit considerations of mitigation measures on acrylamide content of foods–a case study on potatoes, cereals and coffee. British Journal of Nutrition 2008, 99 (S2), S1-S46.
The use of IC50 values to decide on the range of Ochr-A and acrylamide is done correctly; however, under real conditions could they actually be used for a range of toxins that are identical (e.g. 2.3-10um). It is interesting that the combination of the two toxins produced different types of interactions dependent on bioactivity measured, some being additive, synergistic or antagonistic... can the authors explain why these interactions would have different threshold effects.
Response: Thank you for your valuable suggestion. Generally, the co-existence of other compounds with a common pattern of action and/or the same cellular target can result in synergistic or additive interaction. It has been shown that exposure to AA and OTA leads to oxidative DNA damage, which is the main mechanism of cytotoxicity [1-2]. This may explain the synergistic and additive interaction effects we observed in this study. Moreover, the antagonistic effect of AA and OTA may be explained by the competition for glutathione (GSH) in cells. Since electrophiles generated from the metabolism of OTA with Hydroquinone–quinine reduce GSH to produce GSH conjugates, AA can also spontaneously or enzymatically conjugate with GSH to form their corresponding GSH conjugates [4-5]. In practice, the type of interaction between multiple toxins depends on the concentration of toxin used, the duration of exposure, the type of test model chosen, and the indicators assessed.
[1] Nowak, A.; Zakłos-Szyda, M.; Żyżelewicz, D.; Koszucka, A.; Motyl, I. Acrylamide Decreases Cell Viability, and Provides Oxidative Stress, DNA Damage, and Apoptosis in Human Colon Adenocarcinoma Cell Line Caco-2. Molecules 2020, 25, 368.
[2] Mori Y, Kobayashi H, Fujita Y, et al. Mechanism of reactive oxygen species generation and oxidative DNA damage induced by acrylohydroxamic acid, a putative metabolite of acrylamide[J]. Mutation Research/Genetic Toxicology and Environmental Mutagenesis, 2022, 873: 503420.
[3] Faucet‐Marquis V, Pont F, Størmer F C, et al. Evidence of a new dechlorinated ochratoxin A derivative formed in opossum kidney cell cultures after pretreatment by modulators of glutathione pathways: Correlation with DNA-adduct formation[J]. Molecular nutrition & food research, 2006, 50(6): 530-542.
[4] Tozlovanu M, Canadas D, Pfohl-Leszkowicz A, Frenette C, Paugh RJ, Manderville
- Glutathione conjugates of ochratoxin A as biomarkers of exposure. Arh Hig
Rada Toksikol. 2012 Dec;63(4):417-27.
[5] Luo Y S, Long T Y, Chiang S Y, et al. Characterization of primary glutathione conjugates with acrylamide and glycidamide: Toxicokinetic studies in Sprague Dawley rats treated with acrylamide[J]. Chemico-Biological Interactions, 2021, 350: 109701.
Finally, the final message that these results can lead to nutritional decisions needs to be expanded on. What are the decisions that are suggested.
Response: Thank you for your significant reminder. Sorry for the sentence about the nutritional decisions is not quite accurate and we have revised it in the revised manuscript. Please see the Conclusion part.
